# Plant Growth Promotion Using *Bacillus cereus*

**DOI:** 10.3390/ijms24119759

**Published:** 2023-06-05

**Authors:** Iryna Kulkova, Jakub Dobrzyński, Paweł Kowalczyk, Grzegorz Bełżecki, Karol Kramkowski

**Affiliations:** 1Institute of Technology and Life Sciences—National Research Institute, Falenty, 3 Hrabska Avenue, 05-090 Raszyn, Poland; i.kulkova@itp.edu.pl (I.K.); j.dobrzynski@itp.edu.pl (J.D.); 2Department of Animal Nutrition, The Kielanowski Institute of Animal Physiology and Nutrition, Polish Academy of Sciences, Instytucka 3 Str., 05-110 Jabłonna, Poland; g.belzecki@ifzz.pl; 3Department of Physical Chemistry, Medical University of Białystok, Kilińskiego 1 Str., 15-089 Białystok, Poland; kkramk@wp.pl

**Keywords:** direct plant growth promotion, biocontrol, spore-forming bacteria, eco-friendly biostimulants

## Abstract

Plant growth-promoting bacteria (PGPB) appear to be a sensible competitor to conventional fertilization, including mineral fertilizers and chemical plant protection products. Undoubtedly, one of the most interesting bacteria exhibiting plant-stimulating traits is, more widely known as a pathogen, *Bacillus cereus*. To date, several environmentally safe strains of *B. cereus* have been isolated and described, including *B. cereus* WSE01, MEN8, YL6, SA1, ALT1, ERBP, GGBSTD1, AK1, AR156, C1L, and T4S. These strains have been studied under growth chamber, greenhouse, and field conditions and have shown many significant traits, including indole-3-acetic acid (IAA) and aminocyclopropane-1-carboxylic acid (ACC) deaminase production or phosphate solubilization, which allows direct plant growth promotion. It includes an increase in biometrics traits, chemical element content (e.g., N, P, and K), and biologically active substances content or activity, e.g., antioxidant enzymes and total soluble sugar. Hence, *B. cereus* has supported the growth of plant species such as soybean, maize, rice, and wheat. Importantly, some *B. cereus* strains can also promote plant growth under abiotic stresses, including drought, salinity, and heavy metal pollution. In addition, *B. cereus* strains produced extracellular enzymes and antibiotic lipopeptides or triggered induced systemic resistance, which allows indirect stimulation of plant growth. As far as biocontrol is concerned, these PGPB can suppress the development of agriculturally important phytopathogens, including bacterial phytopathogens (e.g., *Pseudomonas syringae*, *Pectobacterium carotovorum*, and *Ralstonia solanacearum*), fungal phytopathogens (e.g., *Fusarium oxysporum*, *Botrytis cinerea*, and *Rhizoctonia solani*), and other phytopathogenic organisms (e.g., *Meloidogyne incognita* (Nematoda) and *Plasmodiophora brassicae* (Protozoa)). In conclusion, it should be noted that there are still few studies on the effectiveness of *B. cereus* under field conditions, particularly, there is a lack of comprehensive analyses comparing the PGP effects of *B. cereus* and mineral fertilizers, which should be reduced in favor of decreasing the use of mineral fertilizers. It is also worth mentioning that there are still very few studies on the impact of *B. cereus* on the indigenous microbiota and its persistence after application to soil. Further studies would help to understand the interactions between *B. cereus* and indigenous microbiota, subsequently contributing to increasing its effectiveness in promoting plant growth.

## 1. Introduction

Progressive agricultural pollution, such as long-term mineral fertilization (with high doses of N) and climate change, which in turn have a huge impact on water quantity and quality, have contributed to the acceleration of the development of environmentally friendly biostimulators and plant protection agents [1,2,3,4,5]. One of the environmentally safe alternatives for chemical agents can be biofertilizers including plant growth-promoting bacteria (PGPB) [6,7]. In sustainable agriculture, PGPB may play an important role by improving yields and soil properties, reducing the growth of phytopathogens, alleviating abiotic and biotic stresses, and enhancing soil biodiversity [8]. PGPB have many beneficial traits that directly or indirectly stimulate plants. Some of them are the auxins (e.g., indole-3-acetic acid (IAA)), cytokinins, and gibberellins production, aminocyclopropane-1-carboxylic acid deaminase (ACC) production, atmospheric nitrogen fixation, phosphorus solubilization, lytic enzymes production (chitinase, cellulase, protease, glucanase), siderophore production, induced systemic resistance (ISR), and antibiotic lipopeptides production (Figure 1) [9,10,11,12,13]. PGPB belong to different genera such as *Acinetobacter*, *Aeromonas*, *Agrobacterium*, *Allorhizobium*, *Arthrobacter*, *Azoarcus*, *Azorhizobium*, *Azospirillum*, *Azotobacter*, *Bacillus*, *Bradyrhizobium*, *Burkholderia*, *Caulobacter*, *Chromobacterium*, *Delftia*, *Enterobacter*, *Flavobacterium*, *Frankia*, *Gluconacetobacter*, *Klebsiella*, *Mesorhizobium*, *Micrococcus*, *Methylobacterium*, *Paenibacillus*, *Pantoea*, *Pseudomonas*, *Rhizobium*, *Serratia*, *Streptomyces*, *Thiobacillus*, and *Variovorax* [14,15,16].

In terms of research, one of the most interesting groups is the *B. cereus* group, which includes at least eight ecologically diverse but phylogenetically highly related species: *B. anthracis*, *B. cereus*, *B. thuringiensis*, *B. mycoides*, *B. pseudomycoides*, *B. weihenstephanensis*, *B. cytotoxicus*, and *B. toyonensis* [17,18]. *B. cereus* species is classified as a gram-positive, aerobic, or facultative anaerobic bacteria, mobile and capable of spore-forming in the presence of oxygen. In the environment, *B. cereus* is a ubiquitous bacterium and can be found in soil, dust, air, water, decaying matter, food products, and plant roots [17]. Some *B. cereus* strains produce several bacterial enterotoxins (HBL, NHE, CytK) that cause two types of food poisoning syndromes in humans and animals, e.g., vomiting and diarrhea [19]. However, non-pathogenic *B. cereus* strains exhibit plant growth-promoting traits and can be used in agriculture as biofertilizers and biocontrol agents [20,21].

This review concisely and comprehensively presents the most relevant information on the direct and indirect mechanisms *B. cereus* strains use, alone or in combination with other PGPB species, to enhance plant growth and help inhibit phytopathogens under stressed and non-stressed conditions. Furthermore, it focuses on how *B. cereus* can colonize and persist in the rhizosphere. The review also addresses the impact of *B. cereus* on the native microbiota of inoculated plants and soil.

## 2. Biofilm Formation in *Bacillus cereus*

*B. cereus* strains colonize plant roots by forming biofilms [22]. Biofilm-forming microorganisms are beneficial to plant health due to increasing resistance to antibiotics, chemicals, heat, UV radiation, and other environmental stresses [23,24]. Some strains of *B. cereus* often merge with other bacterial species, resulting in the formation of mixed biofilms, e.g., two-species biofilms [25]. The *B. cereus* 905 strain isolated from the wheat rhizosphere can exhibit at least two modes of biofilm formation, depending on the environmental conditions. On one hand, *B. cereus* 905 is able to form floating biofilms (pellicles) in a similar way to *Bacillus subtilis*. However, this species can also form biofilm in a manner reminiscent of the biofilm formation using the pathogen *Staphylococcus aureus* [26].

Biofilm formation is associated with various gene expressions. Compared with *B. subtilis*, the regulatory mechanisms that are involved in biofilm formation in environmental *B. cereus* strains are not fully understood. However, recent scientific advances regarding this bacterial species provide more information about the genes and their functions associated with biofilm [27]. Studies conducted on the environmental isolates of *B. cereus* AR156, *B. cereus* 0–9, and *B. cereus* 905 show how biofilm formation is carried out [27,28].

In the environmental isolate *B. cereus* AR156, 23 genes associated with biofilm formation were identified by a random transposon insertion mutagenesis [28]. Importantly, the *ClpYQ* protease gene and genes involved in purine (*purD* and *purH)* and nucleotide biosynthesis—and GTP homeostasis—contribute to biofilm formation and swarm motility in *B. cereus* AR156. Furthermore, *sigB* and several other genes in the putative *SigB* regulon also play important roles in the biofilm formation in *B. cereus* AR156 [28]. It has also been shown that in *B. cereus* AR156, the *comER* gene plays an important role, regulating both biofilm formation and spore formation, and is likely to be a part of a regulatory pathway involved in the activation of Spo0A (the key regulator of biofilm formation and sporulation in *B. cereus* and *B. subtilis*). It has also been reported that the *comER* gene may regulate Spo0A activity through its effect on the small checkpoint protein Sda, which plays an important role in cell development processes in *Bacillus* spp. [29]. Another study has shown that the genes *bcspo0A*, *bcsinI*, and *bcsinR*, regulated by the Spo0A-SinI-SinR regulatory circuit, are essential for sporulation and biofilm formation in *B. cereus* AR156. Interestingly, *bcspo0A* and *bcsinI* genes are important for nematicidal activity and biocontrol against *Meloidogyne incognita* with *B. cereus* AR156 [30].

Importantly, the regulatory protein SpoVG derived from *B. cereus* 0–9 controls the activation of Spo0A transcription and is crucial for both sporulation and biofilm formation [31]. This study has also shown that SpoVG influences AbrB and SinI/SinR networks and thus biofilm development. Moreover, the genes *ptsI*, s*odA1*, *sodA2*, g*apB*, and YmdB protein have been shown to play an important role in biofilm formation in *B. cereus* 0–9 [32,33,34,35]. Importantly, the above-mentioned studies show that the *ptsI* gene may be one of the key genes involved in the control of wheat sharp eyespot using *B. cereus* [32]. Moreover, the *sodA1* gene has a crucial role in spore formation and tolerance to intracellular oxidative stress, while the sodA2 gene plays an important role in the negative regulation of phospholipase and hemolytic activity of *B. cereus* 0–9 [33]. The gapB gene, encoding glyceraldehyde-3-phosphate dehydrogenase (GAPDH), is involved in biofilm formation and extracellular DNA release of *B. cereus* 0–9 by regulating the expression or activity of the LrgAB autolysis regulator [34]. In turn, the YmdB protein is involved in the adaptation of *B. cereus* 0–9 to changing environmental conditions [35]. Interestingly, the *sodA2* gene, encoding manganese-containing superoxide dismutase (MnSOD2), is particularly important in root colonization and biofilm formation in *B. cereus* 905 [36]. In the studies, Gao et al. [36,37,38], using random insertional mutagenesis of the TnYLB-1 transposon, identified the genes *ptsI*, *ptsH*, *recA*, *hrcA*, *clc*, *feoB1*, *feoB2*, *ndk*, and *hutH*, which regulate *sodA2* expression in *B. cereus* 905, and are essential for biofilm formation. 

In conclusion, *B. cereus*, compared with most species of the genus *Bacillus*, has a relatively well-studied mechanism of biofilm formation. Nevertheless, it is still necessary to search for the molecular mechanisms responsible for biofilm formation in order to gain potentially valuable information that could lead to an increase in the effectiveness of *B. cereus* as a PGPB, e.g., by overexpressing certain genes associated with biofilm formation in *B. cereus*.

## 3. *Bacillus cereus* as a Directly Plant Growth Stimulating Bacteria

The main mechanisms of PGPB activity are extensively covered by many review articles [9,39]; thereby, they are not discussed in this review in detail. Instead, it focuses on the role of *B. cereus* strains as plant growth promoters. Due to their ubiquity and close interaction with plants, *B. cereus* strains are often considered as PGPB in various studies [40]. Frequently, a single *B. cereus* strain can promote the growth of many different plant species, indicating that it is well adapted to different habitat conditions. Plant growth stimulation using *B. cereus* strains is measured by parameters such as shoot/root length, fresh/dry biomass, and chlorophyll content. Stimulation of plant growth with *B. cereus* has been documented, e.g., for plants such as soybean (*Glycine max* L. Merr.), wheat (*Triticum aestivum* L.), Chinese cabbage (*Brassica rapa* L., *Chinensis Group*), maize (*Zea mays*), potato (*Solanum tuberosum* L.), pea (*Pisum sativum* L.), and rice (*Oryza sativa* L. var. FARO 44) [20,41,42,43,44].

Most of the studies on direct plant growth promotion with *B. cereus* have been conducted under controlled conditions. For instance, an experiment conducted in in vitro conditions showed that *B. cereus* MEN8 was capable of increasing seed germination parameters (e.g., vigor index, germination energy, or percentage of germination seeds) and stimulating plant growth parameters such as root and shoot length and plant weight of chickpea [45]. Another study in in vitro conditions reported that, after inoculation of rice seeds with *B. cereus* strain GGBSU-1, the germination percentage after 4 days was 100% compared with 65% in the control [44]. Furthermore, after 28 days, bacterial inoculation increased not only morphological parameters and biomass of rice seedlings, but also biochemical parameters, including chlorophyll a and b content, total soluble sugar, and α-amylase [44]. On the other hand, under greenhouse conditions, *B. cereus* T4S, isolated from the sunflower root endosphere by Adeleke et al. [46], promoted sunflower growth, including tap root length, root length, root number, root weight, seed weight, and shoot weight, in comparison with non-inoculated treatment. In addition, the study conducted under pot experiment by Kumar et al. [41] revealed the ability of *B. cereus* LPR2 (isolated from the spinach rhizosphere), alone and in combination with silver nanoparticles (AgNPs), to promote maize plant growth (including root and shoot growth and fresh and dry weight) through phosphate solubilization and IAA, HCN, and ammonia production. Moreover, maize seeds coated with the microbial inoculants *B. cereus* LPR2 and LPR2 with AgNPs exhibited a 37.5 and 25% larger increase in germination, respectively, compared with uninoculated seeds [41]. 

It was also observed that the application of *B. cereus* P8 resulted in a statistically significant increase in plant shoot and root biomass of pea plants under poly-house conditions [43]. This strain showed significant production of IAA and siderophores and had phosphate and potassium solubilization activity [43].

However, there were also studies carried out under field conditions. For instance, Ali et al. [42] showed that potassium solubilizing *B. cereus* was able to increase plant height and shoot dry weight, as well as increase the number of potatoes under field conditions. In addition, the application of this *B. cereus* strain contributed to an increase in total potato yield of approximately 20% compared with non-inoculated plants. Importantly, *B. cereus* also caused an increase in leaf N, P, and K concentrations in inoculated plants compared with the control [42]. In addition, *B. cereus* YL6 promoted soybean and wheat growth in pot experiments, including an increase in leaf total phosphorus [20]. Nevertheless, it also increased the yield of Chinese cabbage under field conditions. Its plant growth-promoting properties were related to IAA and gibberlines production and dissolution of inorganic and organic phosphorus [20]. 

However, despite the fact that *B. cereus* is capable of directly promoting plant growth (from biometric traits to NPK and chlorophyll content or enzyme activity), there is still little research under field conditions, which is an essential step in the commercialization of biofertilizers. 

## 4. Plant Growth Promotion by *Bacillus cereus* in Combination with Other PGPB Species

As is well known, the application of one or more bacterial strains stimulating plant growth can be used as a more rational and safer alternative to agrochemicals [47]. To date, several studies have demonstrated the potential of bacterial consortia with *B. cereus* to improve plant growth. Nevertheless, not much further research is being conducted on the stimulation of plant growth with *B. cereus* in the consortium under controlled conditions. Interestingly, to date, research has shown that *B. cereus* with other bacterial strains (e.g., *Pseudomonas* spp. and *Azospirillum* spp.) can promote the growth of either crops or medicinal plants [48,49,50,51]. For instance, in studies on wheat in a 2-year experiment, the enhancement of plant growth treated in consortium with *B. cereus* (phosphorus-solubilizing microorganisms) has also been reported (under controlled and field conditions) [51]. The consortium containing *B. cereus* (accession no. LN714048) and *Pseudomonas moraviensis* (accession no. LN714047) led to an increase in biometric parameters, including plant height, plant fresh weight, and seeds weight [51]. Chauhan et al. [50], in a 180-day experiment under field conditions, showed a significant increase in plant biomass of turmeric *(Curcuma longa* L.) rhizomes by using a consortium of two bacterial strains: *B. cereus* TSH77 and *B. endophyticus* TSH42. Moreover, the study conducted by Sivasankariv and Anandharaj [48] on the *Vigna unguiculata* plant under in vivo and controlled conditions indicates that the consortium of *B. cereus* GGBSTD1 and *Pseudomonas* spp. GGBSTD3 (isolated from vermisources) acts as a potential biofertilizer due to its plant growth-promoting traits such as phosphate solubilization, IAA production, and siderophores. The results of the study showed that, compared with the control, the consortium significantly enhanced germination percentage, shoot length, root length, leaf area, chlorophyll a and b content in leaves, total chlorophyll content in leaves, fresh weight, and dry weight of plants. As far as chlorophyll is concerned, the positive effects of consortia involving *B. cereus* have already been reported in other studies. For instance, in a study conducted in a greenhouse, after double inoculation with a consortium of strains of *B. cereus* SrAM1 and *Azospirillum brasilense*, *Stevia rebaudiana* seedlings were characterized by an increase in chlorophyll a and chlorophyll b content and total chlorophyll content [49]. In addition, *B. cereus* in combination with other strains can also increase the activity of antioxidant enzymes, including catalase activity [49,51] and significant up-regulation of the ent-KO, UGT85 C, UGT74G1, and UGT76G1 genes responsible for steviol glycoside biosynthesis [49]. Interestingly, in the previously mentioned study by Chauhan et al. [50], a bacterial consortium with *B. cereus* increased the content of the main bioactive component, curcumin, by 13% compared with the control. In summary, *B. cereus* (in combination with other strains) not only promotes plants by increasing their yield and biometrics traits (root length, fresh weight, etc.), but also affects the activity of certain enzymes (e.g., antioxidant enzymes) and increases the concentration of biologically active substances that can be used in medicine, e.g., curcumin.

## 5. *Bacillus cereus* in the Alleviation of Abiotic Stress

Abiotic stress harms the biochemical, morphological, molecular, and physiological functioning of plants by exerting oxidative, osmotic, and ionic stresses. The most common ones are drought, salt, heavy metal pollution, and heat stress, to which crop plants are exposed worldwide [52,53]. Several experiments conducted by different researchers have shown that *B. cereus* strains are capable of promoting plant growth effectively not only in normal conditions but also in stressful environmental conditions [54,55,56,57]. In previous studies, it has been shown that, after inoculation, some plant growth-promoting *B. cereus* strains have considerably beneficial effects on both physiological (dry and fresh weight, shoot and root length, germination) and biochemical traits (chlorophyll content, relative water content, protein, proline, and antioxidant activity) under abiotic stress conditions. Abiotic stress tolerance induced with *B. cereus* strains in plants is mediated by mechanisms such as phytohormone and ACC deaminase activity, antioxidant defense (SOD, POD, APX, CAT, GR), accumulation of osmolytes, volatile compounds (VOCs), and heavy metals bioremediation, including exopolysaccharide production (EPS) and alteration of root morphology [56,58,59,60,61,62].

Recently, in the context of climate change, it has been particularly important to find PGPB effective in mitigating drought and heat stresses. Under drought stress conditions (greenhouse experiment), *B. cereus* UFGRB2 affected the photosynthetic efficiency in soybean by sustaining the potential quantum yield of PSII and maintaining the photosynthesis rate, while a decrease was observed in non-inoculated plants [63]. In another case, the application of *B. cereus* MKA4 under wheat (drought-sensitive variety HD2733) drought stress resulted in a noticeable increase in SOD, catalase, and glutathione reductase (GR) activities compared with control plants under drought conditions (pot experiment) [61]. In addition, co-inoculation of wheat under drought stress (under field conditions) using *B. cereus* (accession CP003187.1) with *Pseudomonas fluorescens* (accession GU198110.1) led to a significant increase in chlorophyll content and improved yield parameters compared to the control [64].

Interestingly, the ACC deaminase-producing *B. cereus* KTES strain was able to enhance shoot and root length, shoot and root weight, and leaf area of *Solanum lycopersicum* under heat stress conditions in the growth chamber [56]. Moreover, isolated by Khan et al. [53], the endophytic strain *B. cereus* SA1 was able to produce IAA, gibberellin, and organic acids. Inoculation of this bacterial strain improved the values of biomass, chlorophyll content, and chlorophyll fluorescence of soybean under standard and heat stress conditions after 10 days of cultivation in the growth chamber (controlled conditions). *B. cereus* SA1 also reduced the levels of heat-stress-produced abscisic acid (ABA) and reduced the amount of salicylic acid (SA).

Importantly, strains of *B. cereus* are also able to enhance plant growth promotion under saline conditions. For instance, isolated from the *Cenchrus ciliaris* (halophytic weed) endophytic strain, *B. cereus* (accession LN714048) caused a significant increase in proline, phytohormones, antioxidant enzymes, and yield parameters such as seed weight and spike length of wheat under saline stress (in vitro conditions) [65]. Interestingly, Zhou et al. [66] have shown that *B. cereus*, through enhancing the activity of antioxidants, increases the growth and photosynthesis ability of cucumber seedlings grown on a solid medium irrigated with 150 mM NaCl solution. On the other hand, Wang et al. [67] conducted a more comprehensive study (pot experiment) focusing on elucidating the mechanisms of action of *B. cereus* in the mitigation of salinity stress in *Glycyrrhiza uralensis*. For instance, *B. cereus* G2 noticeably enhanced proline and glycine betaine content due to upregulated expression of the *BADH1* gene and soluble sugar content due to the activated expression of *α-glucosidase* and *SS* genes, which may lead to a reduction in the osmotic potential of the cell, thereby mitigating osmotic stress. In addition, *B. cereus* G2 mitigated oxidative stress through the effect of antioxidant enzymes [67].

*B*. *cereus* strains are also capable of mitigating plant stress caused by heavy metals, including arsenic, nickel, bo r, cadm, chromium, lead, copper, and zinc [60,68,69,70,71]. Under cadmium stress conditions (pot experiment), *B. cereus* S6D1-105 was able to promote the growth of rice under hydroponic conditions. Cd-tolerant *B. cereus* had the ability to, for example, solubilize potassium and phosphate and produce siderophores, thereby increasing plant biomass, chlorophyll content, and antioxidant enzyme activities, including polyphenol oxidase, catalase, superoxide dismutase, and ascorbate peroxidase and reducing malondialdehyde in either rice leaves or roots [68]. Similar patterns were also obtained by other authors, e.g., Sahile et al. [69] documented that *B. cereus* ALT1 reduces ABA and enhances SA contents in soybean plants under Cd stress conditions (growth chamber conditions). In addition, after plant inoculation of *B. cereus* ALT1, the researchers noted an increase in yield parameters and chlorophyll content. Additionally, *B. cereus* can also alleviate plant stress caused by the presence of chromium in the soil [70]. Application of *B. cereus* contributed to direct *Brassica nigra* plant growth promotion, including increasing shoot and root length and biomass, compared with the uninoculated control. Moreover, inoculation with this strain reduced chromium toxicity (pot experiments, greenhouse). Importantly, after *B. cereus* application, the photosynthetic pigments content was enhanced as well as increases in chlorophyll a, chlorophyll b, and carotenoids were observed. In addition, the strain contributed to an increase in the activity of antioxidant enzymes such as superoxide dismutase, catalase, and peroxidase [70].

Research has also demonstrated the possibility of alleviating plant stress caused by heavy metals by combining *B. cereus* with other bacterial strains. For example, the consortium of three heavy metal-tolerant plant growth-promoting bacteria—*B. cereus* MG257494.1, *Alcaligenes faecalis* MG966440.1, and *Alcaligenes faecalis* MG257493.1—showed tolerance to cadmium, lead, copper, and zinc (in vitro conditions) [71]. The results of the greenhouse study showed that the inoculation resulted in reduced bioaccumulation of heavy metals in the roots and shoots of *Sorghum vulgare* L. 

In conclusion, using some *B. cereus* strains can effectively mitigate the damage caused by abiotic stresses primarily through IAA and ACC deaminase production, increasing antioxidant activity, proline content, and bioremediation of heavy metals.

Tested strains that stimulate plant growth under adverse environmental conditions, such as *B. cereus* (accession CP003187.1), *B. cereus* KTES, *B. cereus* SA1, *B. cereus* (accession LN714048), *B. cereus* G2, *B. cereus* S6D1-105, and *B. cereus* ALT1 alone or in combination with other PGPBs, can be considered for use in developing biofertilizers. Nevertheless, we believe that research with a genetic focus is needed to further understand the mechanisms of B. cereus in stress tolerance in plants.

## 6. Biocontrol Using *Bacillus cereus*

*B. cereus* strains, as endophytic microorganisms and rhizobacteria in interaction with plants, not only promote direct plant growth, but also exhibit interesting traits for biocontrol of plant diseases, which is shown in Figure 2. Some strains of *B. cereus* have shown biocontrol capabilities against various plant pathogens, including bacteria, fungi, oomycetes, nematodes, and protozoan [41,72,73,74,75,76].

*B. cereus* can control bacterial phytopathogens by triggering induced systemic resistance (ISR), which is mediated by the salicylic acid (SA), jasmonic acid (JA), and ethylene (ET) pathways [77]. For instance, *B. cereus* AR156 was able to trigger ISR against *Pseudomonas syringae* pv. tomato DC3000-influenced gene expression (in *Arabidopsis*) by transcription factors WRKY70 and WRKY11, which then affected both the SA and JA/ET signaling pathways [78]. Interestingly, in a subsequent study, Jiang et al. [79] demonstrated that *B. cereus* AR156 also induced ISR against *P. syringae* pv. tomato DC3000, however, by suppressing miR472 and activating CNLs-mediated basal immunity in *Arabidopsis* model system. Recently, Hernández-Huerta et al. [80] demonstrated that *B. cereus* showed similar activity to a commercial agent (*B. subtilis*) against *Xanthomonas euvesicatoria,* the authors suggested that the biocontrol agent may be based on ISR induction. Moreover, *B. cereus* INT1c was able to inhibit *P. syringae* by producing AHL (acyl-homoserine lactone)-lactonase, i.e., an enzyme that breaks down important quorum-sensing (QS) molecules [81]. In addition, endophytic *B. cereus* (strain VT96) was capable of inhibiting QS-regulated virulence factors in *P. aeruginosa* (strain PAO1) and *Pectobacterium carotovorum* by producing AHL-lactonase enzyme (encoded by *aiiA* gene) [82]. Importantly, *B. cereus* was also able to control *Ralstonia solanacearum,* which is an important bacterial plant pathogen as well [72,83]. In the case of the study of Yanti et al. [72], the biocontrol agent involved in *R. solanacearum* suppression was not known. However, Wang et al. [83] suggested that *B. cereus* AR156 induces some specific components in tomato root exudates that may be involved in a decrease in the abundance of *R. solanacearum*.

One of the most important mechanisms against plant pathogens in *B. cereus* is the biosurfactant production of antimicrobial compounds, e.g., ribosomally as well as non-ribosomally synthesized antimicrobial peptides (NRPs) [9,84]. *B. cereus* genomes show a high abundance of NRPs gene clusters, such as cyclic lipopeptides (e.g., iturin A, C, D; fengycin D, CAE; plipastatin A; bacillomycin B, C, D; bacilysin A, D; and surfactin A, C), the dodecapeptides bacitracin, bacillibactin, and hybrid polyketide-nonribosomal peptide zwittermicin A. In *B. cereus* strains, lipopeptides exhibit strong activity against fungal pathogens such as *Fusarium oxysporum*, *Colletotrichum gloesporioides*, *Cercospora lactucae-sativae*, *Alternaria solani*, *Leptosphaeria maculans*, *Sclerotinia sclerotiorum*, *Fusarium solani*, *Leptosphaeria maculans*, *Magnaporthe oryzae*, *Rhizoctonia solani*, and *Sclerotium rolfsii* [46,85,86,87,88,89,90,91,92,93]. For instance, *B. cereus* AK1 isolated from soil was able to synthesize kannurin, which is a lipopeptide antibiotic that shows similar antifungal activity to surfactin, but slightly stronger [94]. Other studies on *B. cereus* strains have also reported the ribosomal synthesis of peptide genes that predominantly encode bacteriocins or bacteriocin-like inhibitory substances (BLIS) such as subtilin, cerein 7A, and cerein 7B [87,95,96]. In addition, as in the case of bacterial phytopathogens biocontrol, PGPB can trigger ISR against pathogenic fungi through the activation of the SA, JA, and ET pathways [97,98,99,100]. However, in contrast to the *B. cereus* AR156-elicited ISR against *P. syringae*, the AR156-triggered ISR against *B. cinerea* does not require the SA signaling pathway [101]. In the case of fungal phytopathogens control, the ISR regulatory pathways are activated by various factors, e.g., recent reports suggest that oxalic acid (OA), secreted by *B. cereus* AR156 through the activation of the JA/ET pathway, provides resistance against the phytopathogenic *Botrytis cinerea* [97]. The JA and SA pathways produced by *B. cereus* EC9 enhance the Kalanchoe immune system against *F. oxysporum* [98]. Furthermore, VOCs produced by *B. cereus* C1L and *B. cereus* YMF3 strains activate the ISR, leading to effective biocontrol against not only *B. cinerea* but also *Cochliobolus heterostrophus* and *Arthrobotrys oligospora* [99]. The role of VOCs in ISR has also been detected in *B. cereus* MH778713 during tomato plant defense against *F. oxysporum* [100]. Another biocontrol mechanism of fungi present in *B. cereus* strains is the activity of lytic enzymes such as cellulases, chitinases, glucanases, chitosanases, and proteases, which efficiently hydrolyze the main components of fungal cell walls [92,102]. Several *B. cereus* strains producing lytic enzymes with antifungal activity are shown in Table 1. 

In addition, strains *B. cereus* KSL-24 and *B. cereus* KSL-8T, which produce HCN, catalase, IAA, and siderophores, may suppress the infection caused by *Phytophthora capsici* belonging to the Oomycetes. It was found that the inoculated seeds caused an inhibition of *P. capsici* mycelial growth in vitro by 75.3 and 88.3%, respectively, compared with the control [74].

Root nematodes (*Meloidogyne* spp.) are also important plant pathogens against which chemical nematicides are often used; however, they also have detrimental effects on human health and the environment [111]. Many studies have proven that *B. cereus* strains exhibit nematocidal activity and have great potential for effective protection of plants against nematode infection [112,113,114,115,116,117,118,119]. *B. cereus* CCBLR15 with biocontrol and plant growth-promoting properties (such as phosphate solubilization, production of siderophores, esterases, gelatinases, and chitinases) was found to have nematocidal activity against *Radopholus duriophilus* in young *Coffea arabica* plants [112]. In turn, Wang et al. [113] documented that *B. cereus* BCM2 caused a significant reduction in the nematode population and increased yield by colonizing tomato roots, competing with *Meloidogyne incognita* for an ecological niche. Moreover, Li et al. [114] observed that the *B. cereus* BCM2 had high activity against second-stage juveniles of *M. incognita* after application on tomato; *B. cereus* BCM2 produced nematode-reducing molecules, e.g., 3,3-dimethyloctane and 2,4-di-tert-butylphenol. In addition, in a recent study on *B. cereus* BCM2, Hu et al. [115] identified chitosanase, an alkaline serine and neutral protease that also contributes to the inhibition of *M. incognita*. Interestingly, research by Xiao et al. [116] reported that *B. cereus* X5 in combination with organic fertilizer, by colonizing tomato rhizosphere soil, significantly reduced the number of tomato root galls and egg masses, and the control efficiency of *Meloidogyne* spp. reached 63.1%. In another study, the *B. cereus* Bc-cm103, by activating the induced immunity of cucumber (*Cucumber sativus*), significantly reduced the number of *M. incognita* galls and reduced the disease index. In addition, *B. cereus* Bc-cm103 formed a biofilm on the surface of cucumber roots and activated the expression of PR1, PR2, LOX1, and CTR1 genes in seedlings, which are markers of SA, JA, and ET signaling pathways [117]. Moreover, Gao et al. 2016 [118], using LC–MS, identified two nematicidal compounds: sphingosine and phytosphingosine, of which sphingosine can induce reactive oxygen species (ROS) in *M. incognita* and thus lead to a lethal effect on nematodes. In addition, nematocidal activity against second-stage larvae of *Heterodera filipjevi* was demonstrated in vitro by treatment with bacterial suspension with a *B. Cereus* 09B18 strain by 83% and under greenhouse and field conditions reduced the number of white females on wheat plants by 75.9 and 43.5%, respectively [119].

Importantly, the role of *B. cereus* MZ-12, isolated from the rhizosphere of pak choi (*Brassica campestris* sp. *chinensis* L.), was described for the first time as a new biocontrol agent of *Plasmodiophora brassicae* —Protozoa [76]. The study showed that after three applications, the MZ-12 strain was able to colonize the rhizosphere of pak choi with subsequent control of the infection by inhibiting zoospores. Nevertheless, there is a need for further research to provide more information and elucidate the control mechanisms of *P. brassicae* by *B. cereus*.

As shown by the results from the text above, *B. cereus* may be involved in the control of many fungi and nematodes, several bacterial phytopathogens, and one member of Protozoa. However, there is still a lack of research on the induction of ISR, and to our knowledge, there have been no studies comparing its effectiveness in the biocontrol of phytopathogens with commercial chemicals.

## 7. Biocontrol Using *Bacillus cereus* in Combination with Other PGPB Species

It is worth noting that, in some cases, plant defense can be more effective and efficient when induced by microbial consortia, compared with the application of a single microbial inoculant. Therefore, considering the ability of some microbial consortiums to trigger induced systemic resistance and biologically control and rebalance the microbial community, there are many opportunities to exploit PGP traits of microbial consortiums not only to effectively increase yield, but also to resist phytopathogens in plant cultivation [120,121,122,123,124]. A number of studies indicate the possibility of controlling fungal phytopathogens, such as *Rhizoctonia solani,* as well as members of the genera *Fusarium* and *Sclerotium.* Naureen et al. [120] evaluated a synergistic action of the bacterial strains *B. cereus* Z2-7, *B. subtilis* SPS2, *Enterobacter* sp. SPR7, and *Aeromonas hydrophilla* in rice (*Oryza sativa*); the consortium increased plant yield and elicited induced systemic resistance against the phytopathogenic *Rhizoctonia solani*. In addition, it increased the levels of chitinases, peroxidases, and glucanases in rice. Moreover, the findings of Kushwaha et al. [93] revealed that, via activation of various biochemical processes during the plant–*Bacillus* interaction, a consortium of microorganisms (*B. cereus* EPP5, *B. amyloliquefaciens* EPP62, *B. subtilis* EPP65) showed strong inhibitory effects on phytopathogenic fungi (including *Rhizoctonia solani*, *Sclerotium rolfsii,* and *Fusarium solani*) and also improved pearl millet (*Pennisetum glaucum*) plant biomass and its shoot and root length. In another study, the synergistic action of consortium members *B. cereus* Rs-MS53 and *Pseudomonas helmanticensis* Sc-B94 resulted in an increased efficacy against the pathogenic fungi *Rhizoctonia solani* and *Sclerotinia sclerotiorum* [124]. The production of microbial volatile organic compounds (mVOCs) with combinations of these strains has been identified as the main mechanism of antagonism.

Yang et al. [122] showed that cotton plants inoculated with a combination of three strains: *B. cereus* AR156, *B. subtilis* SM21, and *Serratia* sp. XY21 (BBS) under greenhouse and field conditions also reduced the abundance of the fungus *Verticillium dahliae* by 86.1 and 76% and improved yields by 49.9 and 13.7%, respectively, compared with the control. In addition, after BBS application under field conditions, soil properties were significantly improved. Then, the results of a 3-year field experiment conducted by Zhang et al. [121] on sweet pepper (*Capsicum annuum* L.) revealed that the application of BBS was particularly effective in controlling the fungus-like *Phytophthora capsici* and increasing the yield of the plant. In addition, after BBS inoculation, the content of total chlorophyll in the leaves, dissolved sugar content, soluble solids, and vitamin C in the fruit were significantly higher in all seasons in comparison with the control plants. Moreover, the inoculation improved soil chemical properties, which could be caused by the alteration of the microbial community induced by BBS. 

Importantly, *B. cereus,* in combination with other strains, is also capable of suppressing the growth of other pathogenic organisms. For instance, in a recent study under field conditions, the potential of the combination of *B. cereus* BT-23, *Lysobacter antiobioticus* 13-6, and *Lysobacter capsici* ZST1-2 was evaluated on plants of Chinese cabbage (*Brassica rapa* subsp. pekinensis) infected by members of Protozoa—*Plasmodiophora brassicae* [123]. The results showed that after three applications, the disease control effect was 65.78% compared with a commercial fungicide. The authors explain this effectiveness by changing the rhizosphere microbiome. 

In conclusion, it is also worth mentioning that merging *B. cereus* with other bacteria may increase the range of biocontrol traits so that the formulation can more effectively suppress fungal phytopathogens. As previously mentioned, the merged production of mVOCs—with *B. cereus* and *Pseudomonas helmanticensis* —has led to the effective control of *R. solani* and *S. sclerotiorum*. Another case confirms that perhaps a mixture with *B. cereus* EPP5, *B. amyloliquefaciens* EPP62, and *B. subtilis EPP65*, in which, for example, EPP5 had a high level of chitinolytic activity and EP65 had a significantly higher level of proteolytic activity—indicating that the strains in the consortium complemented each other in antagonistic activity against *R. solani*, *Sclerotium rolfsii,* and *F. solani* [93].

## 8. *Bacillus cereus* Impact on Native Microbiota

Very important issues concerning the effectiveness of PGBP are their impact on the native microbiota as well as their survival in the soil or, as in the case of endophytes, in plant tissues [125,126,127]. To date, several papers have been published on the effects of *Bacillus* and related bacteria on the microbiota [128,129]; however, to our knowledge, only one has described the *B. cereus* effect on the indigenous microbiota (using Next-Generation Sequencing (NGS)) [130]. The study conducted by Azeem et al. [130] revealed that *B. cereus* with high-temperature biochar significantly increased the abundance of a few important taxa, including Proteobacteria, Actinobacteria, and Firmicutes, and significantly decreased the amount of Acidobacteria, Planctomycetes, Bacteroides, and Gemmatimonadetes in the soil, compared with the treatment with peat alone. Unfortunately, the researchers did not perform a study on untreated control [130]. In the case of other species of the genus *Bacillus*, studies on their impact on the indigenous microbiota have been conducted with, for instance, *B. pumilus* WP8 (using PCR-DDGE) [128], *B. pumilus* TUAT-1 (using NGS) [131], or *B. subtilis* (using NGS) [129].

Importantly, there is a study confirming the colonization of root surface, root tissues, and leaves of Chinese cabbage, soybean, and wheat by plant growth-promoting *B. cereus* (first, it colonized the root surface and hairs, subsequently penetrating the intercellular spaces and vessels, up to the leaves tissues). This phenomenon was documented using *B. cereus* YL6 labeled with green fluorescent protein (GFP) and fluorescence microscopy evaluation [20,132]. However, to our knowledge, there are still no studies describing the survival of *B. cereus* in soil or plant tissues using qPCR, which is considered the best technique to assess this parameter [126]. On the other hand, some interesting studies using qPCR have emerged for other related species [131,133]. For instance, using qPCR, Win et al. [131] detected *B. pumilus* TUAT-1 in the rhizosphere soil (rice) 5 weeks after inoculation. Moreover, Fu et al. [133] revealed that the abundance of *B. amyloliquefaciens* NJN-6 has been stable in the rhizosphere soil (banana) for 3 years after application. 

In conclusion, there is still a need for research into the effects of *B. cereus* on the native microbiota under different conditions (e.g., climatic) and on different plants and soils. A deeper insight into the structure of the bacterial community even at the phylum level—e.g., the abundance of *Acidobacteria,* which are mostly oligotrophs, or *Firmicutes* abundance, which are mostly copiotrophs—gives us a possibility of estimating the direction of biochemical changes following PGPB inoculation. The more such research is performed, the more insights will be established, which may result in the development of appropriate practices in the use of *B. cereus* as a biofertilizer.

## 9. Conclusions

*B. cereus* is capable of direct stimulation of plant growth, including many important crops and even medicinal plants. Its effects on plants have been documented in both biometric traits and the content of biologically active substances. In addition, this review shows that *B. cereus* is capable of controlling a range of phytopathogens, including bacteria, fungi, nematodes, and Protozoa. Hence, *B. cereus* is a promising PGPB species and could be successfully used as a supplement or an alternative to conventional fertilization in the near future, providing solid competition to chemical suppressants of fungal phytopathogens. However, both in the case of direct PGP and biological control, there are still few studies conducted under field conditions, and there are no studies containing a comprehensive analysis of the effectiveness of the bacteria and mineral fertilizers or chemical fungicides. Importantly, such an approach could accelerate the development of future *B. cereus* formulations. Furthermore, there is only one paper dealing with the effects of *B. cereus* on the indigenous microbiota. As mentioned in the section above, understanding the post-inoculation alterations in the native microbiota, especially the structure of the microbial community and the diversity of bacteria, can contribute to a better understanding of the biochemical changes occurring in the soil, which in turn can translate into practical aspects. Additionally, in order to improve the understanding of the response of the microbial community to the introduction of PGBB, such as *B. cereus*, RNA sequencing for metatranscriptome profiling should also be performed, which would give much more information than 16S rRNA gene sequencing. However, these analyses are much less frequent due to the higher costs than 16S gene sequencing. Moreover, to our knowledge, regarding studies on the effects of *B. cereus* on the native microbiota, the post-inoculation metatranscriptome has not yet been studied. Importantly, there is also little information on the tracking of the aforementioned species in soil or plant tissues. Knowledge of the abundance at a certain time after inoculation may be essential to determine the frequency of application of beneficial bacteria. Therefore, it is the study of PGPB persistence in soil that seems to be another key, so these two aspects are crucial in the analysis of PGPB efficacy; thus, scientists should focus on this issue in the near future. 

Lastly, it is also necessary to perform further research focusing on the increased integration of endophytic *B. cereus* species into other crop management practices, which may already be a promising strategy for achieving higher yields in the future. 

## Figures and Tables

**Figure 1 ijms-24-09759-f001:**
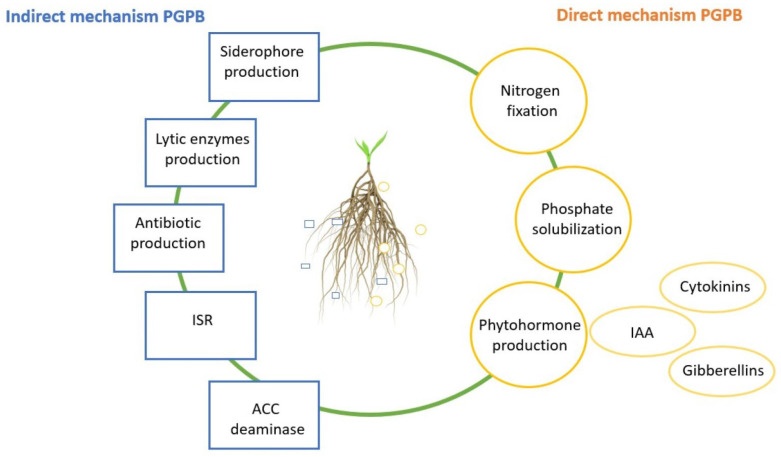
Schematic overview of direct and indirect mechanisms of PGPB.

**Figure 2 ijms-24-09759-f002:**
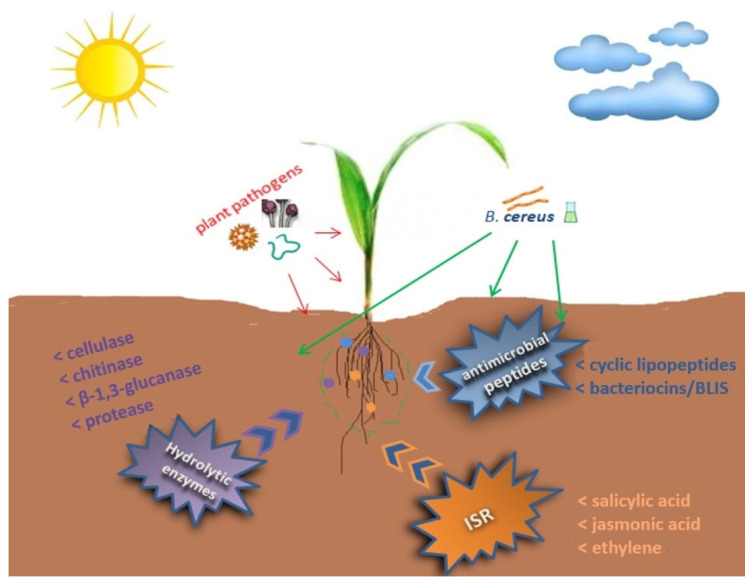
Main mechanisms used by *B. cereus* to combat plant pathogens.

**Table 1 ijms-24-09759-t001:** Biocontrol traits of *Bacillus cereus* strains.

Strains *B. cereus*	Source	Plant Species	Experimental Conditions	Plant Disease and Effects	Lytic Enzymes				References
					Cellulase	Chitinase	β-1,3-Glucanase	Protease	
*B. cereus* YN917	Rice leaf	Rice	Greenhouse	Antifungal activity against *Magnaporthe oryzae*; promoting seed germination and seedling plant growth	+	-	+	+	[103]
*B. cereus* PPB-1	Phyllopla-ne of crop plants	Tomato	In vitro, greenhouse	Antifungal activity against *Fusarium oxysporum, Sclerotium rolfsii, Pythium ultimum*, and *Rhizoctonia solani*	-	+	+	-	[104]
*B. cereus* B25	Rhizosphe-re of maize	Maize	In vitro, in planta, field	Antifungal activity against *Fusarium verticillioides*; increase in grain, yield, plant height	-	+	+	+	[105,106]
*B. cereus* S42	*Tobacco* organs	Tomato	In vitro, in vivo, greenhouse	Antifungal activity against *Fusarium wilt*; increase in height of plants, fresh weight, root length, root fresh.	-	+	-	+	[107]
*B. cereus* IO8	Soil	Tomato	In vitro and in vivo	Antifungal activity against *Alternaria solani, Fusarium solani, Fusarium sambucinum, Alternaria citri, Penicillium occitanis, Aspergillus nidulans, Verticillium dahliae*, and *Botrytis cinerea*	-	+	-	-	[108]
*B. cereus* QQ308	Soil	Chinese cabbage	Pots on the balcony	Antifungal activity against *Fusarium oxysporum, Fusarium solani*, and *Pythium ultimum*; increase in total weight and total height	-	+	-	+	[109]
*B. cereus* CH2	Rhizosphere of eggplant	Eggplant	Greenhouse	Activity against *Verticillium wilt*; increase in plant biomass, both fresh and dry	-	+	-	-	[110]

## Data Availability

Not applicable.

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
