# Peer review of "Plant Growth Promotion Using Bacillus cereus"

_ijms, 2023, doi:10.3390/ijms24119759_

Round 1

Reviewer 1 Report

The manuscript describes the benefits of Bacillus cereus as a bio-inoculant for direct plant growth promotion. A few typos in the text deserve attention when reviewing the proof. The manuscript is relevant to the field and suitable for publication in the IJMS.

The writing is clear, concise, and easy to read.

Author Response

Dear Reviewer 1,

Thank you very much for your favourable review. We have added some more information, corrected the English and improved the references as required by MDPI.

Best Regards

Reviewer 2 Report

Manuscript ID: ijms-2411550

The review work of Kulkova et al. on plant growth promotion by Bacillus cereus is very important for compilation from earlier studies to till date. However, this current version of the review may be improved further for publication standard of the journal.

1.      Abreact needs improvement both in language and content.

2.      Progressive agricultural pollution in the 1st sentence under introduction is not conveying clear meaning. Consider revising the sentences.

3.      Please provide the references for the work, Studies on the environmental isolates of B. cereus AR156, B. cereus 0-9, and B. cereus 905 biofilm formation carried out.

4.      The examples for Bacillus cereus directly as a Plant Growth Stimulating Bacteria in combination with other PGPB Species need to be explained trait-wise and not clubbing together all the traits. Present in a meaningful way starting from prominent earlier works to till the recent work. Provide a clear conclusion for use of the strains as PGPR.

5.      Biofilm formation in Bacillus cereus needs more elaboration and clarification.

6.      Prominent work on biocontrol of Bacillus cereus may clearly be mentioned with examples for bacteria, fungi, oomycetes, nematodes, and protozoan explaining extent of control instead of merging all. Provide a clear conclusion for use of the strains in the sub-section. In addition, compare with the use of chemicals relating to the benefits.

7.      Provide a clear message of biocontrol by Bacillus cereus in combination with other PGPB for its comparative effectiveness and other benefits.

8.      Citation for many reference are missing (e.g. 46). Also, there is no chronology in citing the references e.g 41,45, 43, 47 etc.

9.Conclusion needs improvement by including current status and future thrust.

110.  Gaps need to be corrected in the text (e.g.[ 19 ], [17 ] etc.). Please check whole manuscript.

111.  There is mention of B cereus instead of B. cereus in the text. Please correct.

112.  Improve English language.

Author Response

Dear Reviewer 2,

Thank you for your time and effort in reviewing our manuscript. The feedback has been invaluable in improving the content and presentation of the paper.

We have revised our manuscript according to all comments - new or revised text is marked in blue; the point-by-point responses below.

  1. Abstract needs improvement both in language and content.
  • We have added a few important information and improved the language.
  1. Progressive agricultural pollution in the 1st sentence under introduction is not conveying clear meaning. Consider revising the sentences.
  • We have corrected the sentences.
  1. Please provide the references for the work, Studies on the environmental isolates of B. cereus AR156, B. cereus 0-9, and B. cereus 905 biofilm formation carried out.
  • We have provided the reference.
  1. The examples for Bacillus cereus directly as a Plant Growth Stimulating Bacteria in combination with other PGPB Species need to be explained trait-wise and not clubbing together all the traits. Present in a meaningful way starting from prominent earlier works to till the recent work. Provide a clear conclusion for use of the strains as PGPR.
  • Thank you for your valuable comment, we have structured the text from biometric to biochemical features and summarized the section.
  1. Biofilm formation in Bacillus cereus needs more elaboration and clarification.
  • Thank you for your valuable comment, we have elaborated on several issues and added new information to develop this topic
  1. Prominent work on biocontrol of Bacillus cereus may clearly be mentioned with examples for bacteria, fungi, oomycetes, nematodes, and protozoan explaining extent of control instead of merging all. Provide a clear conclusion for use of the strains in the sub-section. In addition, compare with the use of chemicals relating to the benefits.
  • Thank you for your valuable comment. Following your suggestion we have divided these sections into bacteria, fungi etc. and added new information and conclusions.
  1. Provide a clear message of biocontrol by Bacillus cereus in combination with other PGPB for its comparative effectiveness and other benefits.
  • Thank you for your comment, in the conclusion of the text we wrote about the benefits of merging cereus with other microorganisms in terms of biocontrol efficiency.
  1. Citation for many reference are missing (e.g. 46). Also, there is no chronology in citing the references e.g 41,45, 43, 47 etc.
  • We have corrected it.
  1. Conclusion needs improvement by including current status and future thrust.
  • Thank you for your comment, we have elaborated on the conclusions by developing the issues and adding new information on the direction of future research.
  1. Gaps need to be corrected in the text (e.g.[ 19 ], [17 ] etc.). Please check whole manuscript.
  • We have corrected it.
  1. There is mention of B cereus instead of B. cereus in the text. Please correct.
  • We have corrected it.
  1. Improve English language.
  • We have corrected it.

Reviewer 3 Report

This manuscript is well done and can be expected to make a significant contribution to those areas of microbiology that investigate the symbiotic relationship between a plant host and its microbes.

However, the reviewer has several comments regarding the manuscript.

1. The reviewer believes that this manuscript would have been improved by having a summary or conclusion at the end of each section of the manuscript. The presence of such short conclusions would allow a better understanding of the global idea that the authors were inspired by when writing this manuscript. And, perhaps, section 9 (Conclusions) will improve and become more understandable from this.

2. The Conclusions section looks very limited. The authors analyzed many studies on the interaction of PGPB with plants and their microbiota; conclusions, however, are almost non-existent. The reviewer believes that the authors here have much room for improvement in this section, especially since there is some similarity between this manuscript and the first probiotic reviews published a long time ago, where similar issues were raised.

3. Figure 2 is very poorly readable, as the authors used fonts with low contrast relative to the background/filling of the figure. Apparently, figure 2 should be somehow modified so that the signatures on it (for example, protease or cellulase) are well readable.

In general, the reviewer does not consider the comments on points 1 and 2 particularly critical, but figure 2 (and maybe figure 1) should be made more contrasting.

Author Response

Dear Reviewer 3,

Thank you for your time and effort in reviewing our manuscript. The feedback has been invaluable in improving the content and presentation of the paper.

We have revised our manuscript according to all comments - due to the fact that we have made a lot of corrections to the article, we apologize for the slight confusion in the original text and inform you that the new information has been pasted and highlighted in blue; the point-by-point responses below.

  1. The reviewer believes that this manuscript would have been improved by having a summary or conclusion at the end of each section of the manuscript. The presence of such short conclusions would allow a better understanding of the global idea that the authors were inspired by when writing this manuscript. And, perhaps, section 9 (Conclusions) will improve and become more understandable from this.
  2. The Conclusions section looks very limited. The authors analyzed many studies on the interaction of PGPB with plants and their microbiota; conclusions, however, are almost non-existent. The reviewer believes that the authors here have much room for improvement in this section, especially since there is some similarity between this manuscript and the first probiotic reviews published a long time ago, where similar issues were raised.
  • Thank you for your valuable comment. We have added short summaries to each section and improved the conclusions (section 9) by adding new information on the direction of future research.
  1. Figure 2 is very poorly readable, as the authors used fonts with low contrast relative to the background/filling of the figure. Apparently, figure 2 should be somehow modified so that the signatures on it (for example, protease or cellulase) are well readable.
  • We have corrected the figure 2.

Round 2

Reviewer 2 Report

The current version of the manuscript is improved one than the previous version. This version may be accepted for publication.